# Water tracks intensify surface energy and mass exchange in the Antarctic McMurdo Dry Valleys

Tobias Linhardt[1], Joseph S. Levy[2], and Christoph K. Thomas[1,3]

[1]Micrometeorology Group, University of Bayreuth, Bayreuth, Bavaria, Germany
[2]Department of Geology, Colgate University, Hamilton, New York, USA
[3]Bayreuth Center of Ecology and Environmental Research (BayCEER), University of Bayreuth, Bayreuth, Bavaria, Germany

**Correspondence:** Tobias Linhardt (tobias.linhardt@uni-bayreuth.de)

**Abstract.** The hydrologic cycle in the Antarctic McMurdo Dry Valleys (MDV) is mainly controlled by surface energy balance. Water tracks are channel-shaped high moisture zones in the active layer of permafrost soils and are important solute and water pathways in the MDV. We evaluated the hypothesis that water tracks alter the surface energy balance in this dry, cold and ice-sheet-free environment during summer warming and may therefore be an increasingly important hydrologic feature in the

MDV in the face of landscape response to climate change. The surface energy balance was measured for one water track and two off-track reference locations in Taylor Valley over 26 days of the Antarctic summer of 2012–2013. Turbulent atmospheric fluxes of sensible heat and evaporation were observed using the eddy-covariance method in combination with flux footprint modeling, which was the first application of this technique in the MDV. Soil heat fluxes were analyzed by measuring the heat storage change in the thawed layer and approximating soil heat flux at ice table depth by surface energy balance residuals.

For both water track and reference locations over 50 % of net radiation was transferred to sensible heat exchange, about 30 % to melting the seasonally thawed layer, and the remainder to evaporation. The net energy flux in the thawed layer was zero. For the water track location, evaporation was increased by a factor of 3.0 relative to the reference locations, ground heat fluxes by 1.4, and net radiation by 1.1, while sensible heat fluxes were reduced down to 0.7.

Expecting a positive snow and ground ice melt response to climate change in the MDV, we entertained a realistic climate

change response scenario in which a doubling of the land cover fraction of water tracks increases the evaporation from soil surfaces in lower Taylor Valley in summer by 6 % to 0.36 mm d$^{-1}$. Possible climate change pathways leading to this change in landscape are discussed. Considering our results, an expansion of water track area would make new soil habitats accessible, alter soil habitat suitability and possibly increase biological activity in the MDV.

In summary, we show that the surface energy balance of water tracks distinctly differs from that of the dominant dry soils in

polar deserts. With an expected increase in area covered by water tracks, our findings have implications for hydrology and soil ecosystems across terrestrial Antarctica.

# 1   Introduction

The McMurdo Dry Valleys (MDV) of southern Victoria Land are the largest ice-sheet-free region in continental Antarctica covering a total area of 22,700 $km^2$ and an ice-free area of 4,500 $km^2$ (Levy, 2013). The MDV are characterized by bare permafrost-dominated soils, glaciers, ice-covered lakes and ephemeral streams (Gooseff et al., 2011; Lyons et al., 2000).

Despite their geographical remoteness, the MDV are subject to a changing climate showing inconsistent trends in sign and of varying magnitude over the past decades. From 1986 to 2002 the MDV experienced a cooling trend of 0.7 K per decade (Doran et al., 2002). The cooling stopped around 2002, when high temperatures and insolation caused strong glacial melt and permafrost thawing which led to several well-documented persisting ecosystem and landscape changes including lake level rise (Barrett et al., 2008; Gooseff et al., 2011), increased biological production (Gooseff et al., 2017), increased nematode

abundance (Barrett et al., 2008), increased thermokarst formation modifying stream biogeochemistry (Gooseff et al., 2016; Levy et al., 2013) and increased incision of streams (Fountain et al., 2014). The period post 2002 has shown no significant trends in temperature and a constant high insolation (Gooseff et al., 2017). For the near future, a rise in both temperature and precipitation is predicted throughout Antarctica (Christensen et al., 2013), with a temperature increase by several degrees expected for all seasons in the MDV (Walsh, 2009). Empirical evidence for this new trend is provided by an accelerating

decrease of the Antarctic ice sheet volume (IPCC, 2014).

The hydrologic cycle in the MDV is mainly controlled by surface energy balance (SEB) instead of precipitation, because liquid water is only available in case of net energy uptake at the surface which only occurs in summer (Gooseff et al., 2011). In the summer season, streams connecting glaciers to lakes represent the dominant water flow path in the MDV (Doran et al., 2008; Gooseff et al., 2011). While soils outside the direct vicinity of lakes and streams are generally dry desert soils, a considerable

area of 5 to 10 $km^2$ in the MDV consists of wetted soils in summer (Langford et al., 2015). The water source for these wetted soils can be melting of snow patches, melting of pore ice or buried segregation ice (Harris et al., 2007), and even soil salt deliquescence (Levy et al., 2012).

Water tracks are one widespread type of wetted soils in the MDV, along with seeps and wet patches (Langford et al., 2015). Water tracks are channel-shaped zones of high soil moisture and commonly high salinity in the active layer of permafrost-

dominated soils that form along linear depressions in the ice table as a result of shallow downhill groundwater flow (Gooseff et al., 2013; Hastings et al., 1989; Levy et al., 2011; McNamara et al., 1999). The ice table is defined as the interface between the unfrozen part of the soil column and the underlying frozen part. Although water tracks are small-scale, scattered features in the MDV landscape occupying a small fraction of the total area with widths of ≈1 to 10 m, they represent an important pathway of solute transport; they also show stark contrasting surface radiative, thermal, and soil properties (Langford et al.,

2015; Levy et al., 2011).

Compared to their surroundings, water tracks feature an increased soil water content and, in most cases, elevated solute concentrations (Levy et al., 2011, 2014; Ball and Levy, 2015). Thickness of the active layer is elevated compared to the surrounding soil owing to an increased thermal conductivity and reduced albedo (Gooseff et al., 2013; Ikard et al., 2009; Levy et al., 2011; Levy and Schmidt, 2016). Although water availability is the primary limiting factor to biological activity in the

MDV (Barrett et al., 2007; Kennedy, 1993), water tracks generally show lower biomass and biological activity with highly specialized biological communities which is due to strongly increased salinity that counteracts the beneficial effect of high soil moisture (Ball and Levy, 2015; Levy et al., 2011; Zeglin et al., 2009). The presence of episodic overland flow reduces salinity by washing out solutes which can then lead to increased respiratory activity in water tracks compared to their surroundings (Ball and Levy, 2015).

With SEB being the main control for hydrological processes in the MDV (Gooseff et al., 2011) and with the absence of vegetation cover, soils in the MDV serve as a natural laboratory for investigating the effects of soil hydrology on physical soil properties and processes by means of SEB measurements on surfaces varying in soil water content. Observations and projections of climate change and climate change responses point towards a higher water availability in the MDV caused by increased melting of snow and ground ice (Fountain et al., 2014; Gooseff et al., 2017; Guglielmin and Cannone, 2012; Levy et al., 2013; Walsh, 2009). The resulting soil moisture increase is expected to cause positive feedbacks on soil thawing by enhanced thermal conductivity and energy uptake (Gooseff et al., 2013; Ikard et al., 2009; Levy and Schmidt, 2016). These expected climate change responses may lead to a dramatic increase in abundance and connectivity of wetted soils in the MDV (Ball et al., 2011; Wall, 2007). Thus, the investigation of hydrologic properties and energy and matter exchange of water tracks, as opposed to dry soils, may provide a baseline for energy and matter exchange of soils in the MDV in a warmer, wetter future.

The objective of our study is to quantify the impact of water tracks on vertical energy and water exchange of cold desert soils in the MDV during peak summer warming. Given the lower albedo and higher soil water content of water tracks, we hypothesize that the energy uptake of water track surfaces is enhanced compared to the adjacent, dry, off-track soils. As a result, we expect the latent evaporative and ground heat fluxes to be increased for water tracks compared to their surroundings.

## 2 Materials and methods

### 2.1 Constructing a conceptual surface energy budget

The SEB is defined as the equilibrium between net radiation $Q_S^*$ at the surface and the sum of sensible heat flux $Q_H$, latent heat flux $Q_{LE}$ and ground heat flux $Q_G$:

$$-Q_S^* = Q_H + Q_{LE} + Q_G. \tag{1}$$

The change of heat storage at the surface was neglected, as no vegetation or built infrastructure is present in this environment. In the summer season, we expect evaporation to be the dominating phase change of ice in valley floors generating atmospheric latent heat flux, which is why we defined $Q_{LE}$ as the evaporative heat flux. The magnitude of the ground heat flux in a cold-desert energy budget can be substantial because thawing of ice beneath the ice table – the upper boundary of currently frozen soil – requires a large amount of energy (Lloyd et al., 2001). We divided the ground heat flux $Q_G$ into the temporal change in heat storage in the thawed layer $\Delta S_{TL}$ and into the soil heat flux $Q_{IT}$ at ice table depth which consists of the latent and

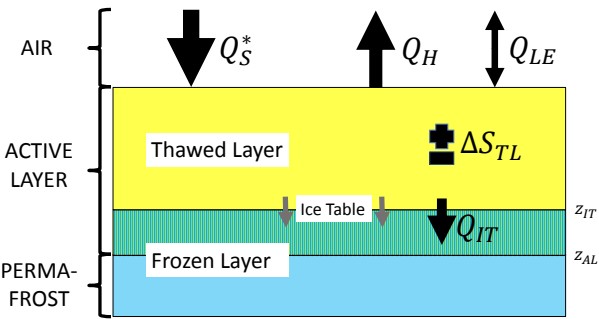

**Figure 1.** Surface energy balance concept used in this study, with net radiation $Q_S^*$, sensible heat flux $Q_H$, latent evaporative heat flux $Q_{LE}$, temporal storage change in the thawed layer $\Delta S_{TL}$ and soil heat flux at ice table depth $Q_{IT}$. Ground heat flux is given by adding together $\Delta S_{TL}$ and $Q_{IT}$. Black arrows depict observed directions of energy fluxes. The soil profile can be divided by the active layer depth $z_{AL}$ into the seasonally thawed active layer and the perenially frozen permafrost, or by the ice table depth $z_{IT}$ into the currently thawed and frozen layers, where the ice table equals the thawing front advancing into the soil, denoted by grey arrows.

sensible heat fluxes into the frozen layer beneath the ice table (Fig. 1):

$$Q_G = \Delta S_{TL} + Q_{IT}, \tag{2}$$

where $\Delta S_{TL}$ equals the temporal heat storage change between surface and ice table depth $z_{IT}(t)$ and was determined using a calorimetric approach following Liebethal and Foken (2007):

$$5 \quad \Delta S_{TL} = \int\limits_{z_{IT}(t)}^{0} \frac{\delta}{\delta t} C_G(z) T(z)\, dz, \tag{3}$$

where $C_G(z)$ $(\mathrm{J\,m^{-3}\,K^{-1}})$ is the volumetric heat capacity of the soil and $z$ denominates depth in the soil.

The turbulent sensible heat flux and latent evaporative heat flux were computed using the eddy-covariance method in combination with flux footprint modeling. The micrometeorological eddy-covariance method is a tool for quantifying turbulent energy and mass net exchange between the land surface and the atmosphere with high precision, and is used at more than 700 sites globally across all climate zones and biomes (e.g. Baldocchi, 2003; Aubinet et al., 1999). It is based upon turbulence observations and the application of the Reynolds decomposition ($X = \overline{X} + X'$), which divides a scalar or vector quantity $X$ into its temporal mean $\overline{X}$ and temporal perturbation $X'$ to compute the net flux. According to Reynolds' second postulate, the total vertical net flux $\overline{wX}$, which is computed from the measurements of the vertical wind speed $w$ and a quantity $X$, can be calculated as

$$15 \quad \overline{wX} = \overline{w}\,\overline{X} + \overline{w'X'}, \tag{4}$$

where the covariance $\overline{w'X'}$ is the turbulent flux. Averaged over a sufficiently long period, the mean vertical wind speed $\overline{w} = 0\,\mathrm{m\,s^{-1}}$ in the surface layer of the atmosphere. Hence, Eq. (4) is simplified to

$$\overline{wX} = \overline{w'X'}. \tag{5}$$

Thus, mean vertical fluxes of scalars in the surface layer can be expressed as turbulent fluxes via covariances of $w$ and a scalar specifying the flux. For the computation of $Q_H$ and $Q_{LE}$, the covariance $\overline{w'T'}$ $(\mathrm{m\,K\,s^{-1}})$ of $w$ and air temperature $T$ and the covariance $\overline{w'q'}$ $(\mathrm{kg\,kg^{-1}\,m\,s^{-1}})$ of $w$ and specific humidity $q$ were used, respectively, to calculate heat fluxes in energetic units $(\mathrm{W\,m^{-2}})$ with Eq. (6) and Eq. (7):

$$Q_H = c_p\rho\overline{w'T'} \tag{6}$$

$$Q_{LE} = \rho\lambda\overline{w'q'}, \tag{7}$$

where $c_p$ $(\mathrm{J\,K^{-1}\,kg^{-1}})$ is the specific heat capacity of air, $\rho$ $(\mathrm{kg\,m^{-3}})$ is the air density, and $\lambda$ $(\mathrm{J\,kg^{-1}})$ is the latent heat of vaporization. Note that we used the latent heat of vaporization and not that of sublimation assuming that the water evaporating from the wetted water track soil is liquid already. The complete SEB equation applied in this study then equates to

$$-Q_S^* = c_p\rho\overline{w'T'} + \rho\lambda\overline{w'q'} + \int_{z_{IT}(t)}^{0} \frac{\delta}{\delta t}C_G(z)T(z)\,dz + Q_{IT}. \tag{8}$$

Flux footprint modeling was used to distinguish between turbulent fluxes originating from water tracks and those from non-water track surfaces. Our intention was to isolate the effect of these small-scale, linear features, which may occupy only several tens to hundreds of square meters, onto the SEB. Flux footprint modeling allows for connecting the recorded turbulence signals at sensor location to their source area (see e.g. Leclerc and Foken, 2014, and references therein). Based upon well-known laws for isotropic and homogeneous turbulent airflows, flux footprint models compute a spatially explicit probability density function, i.e., the flux footprint, quantifying the contribution of each grid cell of the land cover matrix to the total observed flux. Since these contributions and thus the source area vary with sensor height, surface properties, and airflow properties including turbulence statistics, wind speed, and wind direction, one may select intervals for which the turbulent flux predominantly originates from a certain land cover type of interest.

## 2.2 Field observations

The study was conducted in Taylor Valley, a polar desert with 3 to 50 $\mathrm{mm}$ annual precipitation (Fountain et al., 2009) and -18°C mean annual temperature (Doran, 2002). Continuous permafrost soils in lower Taylor Valley show active layer depths between 45 to 75 $\mathrm{cm}$ (Bockheim et al., 2007) and are inhabited by a simple, nematode-dominated soil ecosystem (Priscu, 1998).

**Table 1.** Installation notes for the three eddy-covariance stations, with height of anemometer $z_{an}$, height of infrared gas analyzer $z_{irga}$, horizontal distance between both devices $d_{an,irga}$ and estimated roughness length $z_0$; all lengths in m. PLD and GT are non-water track reference surfaces, and were summarized as NWT.

| Site name | Coordinates | Recording period | Land cover class | $z_{an}$ | $z_{irga}$ | $d_{an,irga}$ | $z_0$ |
|---|---|---|---|---|---|---|---|
| WT | 77.57655° S, 163.48328° E | 26 December 2012–21 January 2013 | water track | 2.04 | 1.98 | 0.19 | 0.03 |
| PLD | 77.58083° S, 163.49234° E | 27 December 2012–14 January 2013 | paleolake delta | 2.01 | 1.96 | 0.17 | 0.01 |
| GT | 77.57925° S, 163.47504° E | 14 January 2013–21 January 2013 | glacial till | 2.01 | 1.96 | 0.17 | 0.03 |

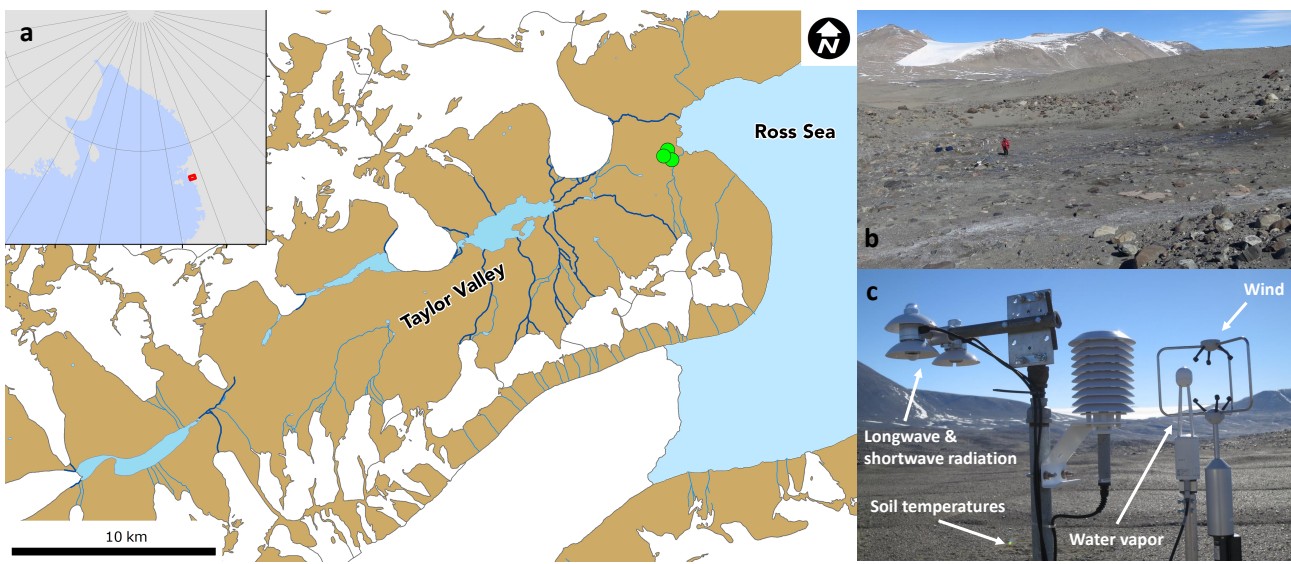

**Figure 2.** Location of the three study sites near Ross Ice Shelf in lower Taylor Vallley, with inset map showing location of the sites on the Ross Sea, gridded by 10° latitude/ longitude lines **(a)**. WT site (water track) **(b)**. PLD site (paleolake delta) with eddy-covariance/ surface energy balance station **(c)**.

Measurements were taken over 26 days under summer conditions from 26 December 2012 to 21 January 2013 at three closely collocated sites near the Ross Sea shore in the valley floor of lower Taylor Valley (Fig. 2a, Tab. 1), with slopes $< 7°$ (supplementary Fig. 1). At any time during the experiment, two eddy-covariance and SEB stations were operated: One station was installed throughout the whole period at the investigated water track (Fig. 2b), referred to as WT. The other station was operated as a reference representing the dominant non-water track, bare soil surfaces in lower Taylor Valley; it was successively installed at two sites with different soil textures: PLD was located on a paleolake delta dominated by fine surficial sediments (Fig. 2c), while GT represented coarse glacial till (Tab, 1, Fig. 3). PLD and GT combined together are referred to as non-water track reference NWT.

Each station consisted of a net radiometer (*NR01, Hukseflux Thermal Sensors B.V., Delft, NL*) for measuring incoming and outgoing longwave and shortwave radiation components, and an ultrasonic anemometer (*81000 VRE, R.M. Young Company, Traverse City, MI, USA*) in combination with an infrared gas analyzer (*LI-7500, LI-COR Inc., Lincoln, NE, USA*) for eddy-covariance measurements (Fig. 2c). Sonic anemometer measurements providing wind and acoustic temperature data, and infrared gas analyzer measurements of water vapor were sampled and recorded at 20 Hz (Tab. 1).

Soil temperatures were recorded in several depths in the thawed layer with thermistors and thermocouples (Tab. C1). Ice table depths were determined at WT and PLD by depth-to-refusal measurements. We measured volumetric heat capacity $C_G$ of the soil in-situ with a thermal properties analyzer (*KD2 Pro, Decagon Devices, Pullman, WA, USA*), averaged from 4 and 31 soil samples from the surface at WT and PLD, respectively, along with thermal conductivity $K_G$ and thermal diffusivity $D_G$.

## 2.3 Calculating surface energy balance components

Eddy-covariance fluxes were computed using a fixed perturbation time scale of 30 minutes using the bmmflux tool software of the Micrometeorology Group of the University of Bayreuth (see appendix in Thomas et al., 2009). First the raw data were filtered by instrument flags and plausibility limits. A despiking routine was applied to exclude unphysical turbulence data (Vickers and Mahrt, 1997). Time lags between gas analyzer and anemometer were corrected by maximizing the covariances of the measured quantities. A three-dimensional rotation routine was used to rotate the flow into the mean streamlines and eliminate the mean vertical wind potentially caused by either a tilt in the sonic anemometer, surface conditions, or semi-stationary eddies of time scales exceeding the perturbations time scale (Wilczak et al., 2001). Computed fluxes were corrected for low- and high-pass filtering following Moore (1986). The buoyancy flux was converted into sensible heat flux by a post-hoc buoyancy correction (Liu et al., 2001). A post-hoc density correction was applied to the latent evaporative heat flux (Webb et al., 1980). Eddy-covariance quality flags for turbulent fluxes following Foken et al. (2004) were used to filter out intervals in which the assumption of stationarity and well-developed turbulence were not satisfied. The scheme runs from 1 (best quality) to 9 (worst quality) and we discarded data with flags $\geq 7$.

$\Delta S_{TL}$ was determined as temporal heat storage change via calorimetry with Eq. (3), from soil temperature profiles which were logarithmically interpolated between the measurements with increments of 0.01 m. The thickness of the thawed layer was assumed to be constant in SEB calculations throughout the measuring period, although in reality it likely changed over time due to lowering of the ice table as a result of continued soil thawing in the active layer which is typically not fully thawed before mid-January to early February in Taylor Valley (Adlam et al., 2010; Conovitz et al., 2006).

For WT, temperature at ice table depth was assumed to be 0 °C. At PLD ground temperature was recorded at the initially measured ice table depth of 0.3 m with a mean of 1.9 °C. We decided to fill gaps in the ground temperatures at ice table depth for NWT with this mean value which was the case for the entire record at GT. Thereby we avoided unphysical spikes in the calculated $\Delta S_{TL}$ for NWT which would have resulted from setting the temperature for these cases to 0 °C at ice table depth.

Despite the simplified assumption of constant ice table depth in SEB calculations and a deviation of temperature at ice table depth from 0 °C for PLD, we argue that the possible error in $\Delta S_{TL}$ is small compared to the flux magnitudes (see Appendix A),

since most of the energy storage change occurred close to the ground surface. The computed $\Delta S_{TL}$ was discarded when calculated from temperature measurements at one depth only, which applied to 34 % of the data for WT.

The second component of the ground heat flux, $Q_{IT}$, was approximated as the SEB residual after quantifying $Q_S^*$, $Q_H$, $Q_{LE}$ and $\Delta S_{TL}$ according to the methods laid out above. Estimating $Q_{IT}$ as residual assumes the SEB to be closed and therefore poses limitations to its interpretation as the conductive energy transported away from the surface into the ground, available to deepen the thawed layer by moving the ice-table depth further into the ground toward the depth of the permafrost layer as the summer season progresses.

Almost all experimental SEB studies in non-permafrost environments using direct flux measurements from EC have reported on a significant residual on the order of 10 to 30 % of the net radiation $Q_S^*$ (see e.g. list in Foken, 2008) with a few, rare exceptions (e.g. Mauder and Foken, 2006). The proposed explanations for the observed systematic residual are manifold and include, after compensating for measurement artifacts and impacts of post-field data processing, surface heterogeneity creating semi-stationary convective structures which are systematically neglected in near-surface observations. We intentionally did not consider applying any of the common methods to eliminate the SEB residual as the results from non-permafrost surfaces may not be applicable to permafrost-dominated ecosystems including the MDV.

While the discussion of the potential causes for the SEB residual is outside of the scope of our study, its implications for estimating $Q_{IT}$ need to be considered. Any systematic observational residual $\epsilon \geq 0 \, \mathrm{Wm}^{-2}$ would systematically enhance $Q_{IT}$ as $\epsilon + Q_{IT} = -Q_S^* - Q_H - Q_{LE} - \Delta S_{TL}$, potentially even beyond physically reasonable limits, and lead to a much deeper thawed layer. We recall that for our MDV sites the energy flux directed toward deepening the melt-front may be the dominant term or of similar magnitude compared to the sensible and evaporative heat fluxes. Hence, significant errors should not go undetected and some certainty if estimating $Q_{IT}$ as the SEB residual should be gleaned from comparing the magnitude of all SEB components, their signs, their relative timing, and by comparing the observed depth to refusal, DTR, to that computed from $Q_{IT}$ assuming a certain ice concentration in the soil. This evaluation is included in the discussion below.

To date, there is no experimental technique based on direct flux measurements which can eliminate the observational residual with certainty. The residual was found to be greatly reduced when using indirect techniques such as large-aperture scintillometry based upon similarity theory or large eddy simulation modeling to estimate SEB components (Foken et al., 2010). For comparison, any SEB model would by definition assume the SEB to be closed, and models are routinely constrained by observations potentially suffering from the observational residual. We therefore do not consider our approach to estimate $Q_{IT}$ a flaw, but rather an assumption which needs to be evaluated against the background of the results and our understanding of the physical heat transfer and melting processes.

The expected lag of the diurnal temperature signal $t_{\mathrm{lag}}$ penetrating from the surface ground to the ice table depth $z_{IT}$ was calculated after Woo (2012) as:

$$t_{\mathrm{lag}} = 0.5 z_{IT} (L \, \pi^{-1} \, D_G^{-1})^{0.5}, \tag{9}$$

with period $L = 1$ day and thermal diffusivity $D_G$.

All SEB components were filtered for physically implausible values. The resulting data gaps corresponded between 2 and 34 % of available data for SEB components. The gaps were filled by linear interpolation whenever possible, leading to a data coverage between 66 and 100 % for WT and 93 to 96 % for NWT (see also supplementary Fig. 2).

## 2.4 Identifying turbulent heat fluxes from the water track via footprint modeling

The time-variant footprint of the turbulent energy and mass fluxes was modeled for WT with the Lagrangian-statistics backward TERRAFEX model of the University of Bayreuth (Göckede, 2001). The land cover matrix needed for footprint modeling was generated by mapping the main land cover classes on a Quickbird satellite image. The used land cover classes include modern stream channels, water tracks, paleolake delta sediments, glacial tills, and exposed ice. Pixel classification was done on the basis of albedo, texture, and field descriptions. Water tracks were mapped as multi-segment lines and were assigned a constant width of 10 m derived from the in-situ visual observations. Stream channels were mapped as multi-segment lines and were assigned a fixed width of 20 m. Landscape regions were given a single classification code avoiding any overlap. Features were mapped to provide continuous plan-view coverage with no gaps between features and no unassigned cells. Vector landscape features were rasterized at $10 \, \mathrm{m \, px^{-1}}$ and were exported into a local Lambert Conformal Conic projection to produce gridded land cover values. Each class was assigned an adequate momentum surface roughness length from literature and used in footprint calculations.

Since flux footprint modeling can provide an accurate estimate of the shape, size, and orientation of the source area, we selected the turbulent sensible and latent evaporative heat fluxes of the water track at WT if a cumulative percentage of at least $\approx 80 \, \%$ was contributed by water track pixels as indicated by the land cover map, which corresponds to $\approx 50 \, \%$ of the footprint spatial extent (Fig. 3). As there was a good match between the size of flux source area and the surface area of the water track, this spatiotemporal filtering is expected to isolate the contribution of the water track at WT to the SEB in a meaningful fashion. Lending to the significant insolation throughout the diurnal cycle in the Antarctic summer in combination with the low albedo of the MDV surface, net radiation was consistently negative showing a net energy gain at the surface. Hence, no dynamically stable atmospheric conditions were observed which would have resulted in increased flux footprint sizes and complicated selection of meaningful heat fluxes dominated by water tracks. Therefore, no further filtering was necessary to avoid errors in turbulent heat fluxes caused by weak turbulence in case of dynamic stability.

## 2.5 Statistics

Average diurnal variations were calculated for all SEB components with all representative data for comparing individual energy fluxes between WT and NWT (see supplementary Fig. 2). We used linear regression models to compare magnitudes of these average diurnal variations between WT and NWT.

For SEB evaluation average diurnal variations were only calculated from data where all SEB components were available, which was fulfilled for 36 % of the data set (see supplementary Fig. 2). Average energy fluxes for the complete measuring period were computed by averaging mean diurnal variations of the SEB, since the data coverage for SEB was unevenly distributed across the diurnal course.

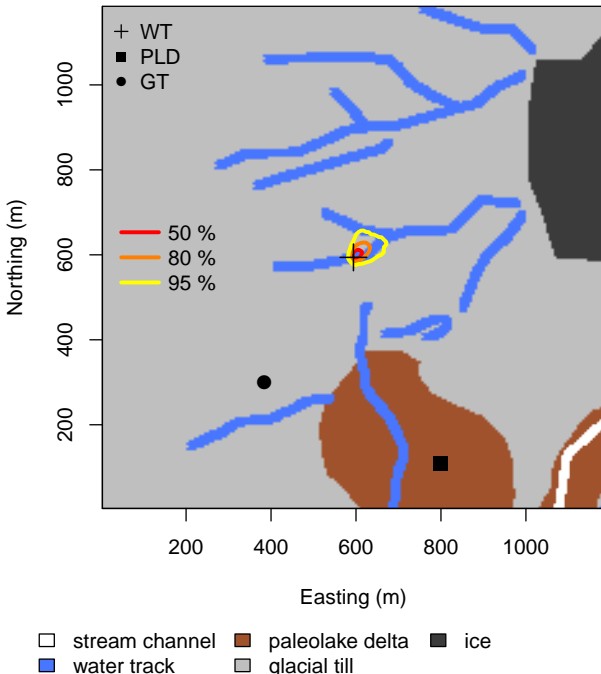

**Figure 3.** Excerpt of the land cover matrix for lower Taylor Valley with eddy-covariance station locations on water track (WT), paleolake delta (PLD) and glacial till (GT) surfaces. Extent of the flux footprint of WT is indicated by three probability density lines.

We estimated average energy fluxes for each SEB component $Q_x$ spatially integrated for bare soil in the entire lower Taylor Valley with

$$Q_{x,\text{TV}} = r_{\text{WT}}Q_{x,\text{WT}} + (r_{\text{PLD}} + r_{\text{GT}})Q_{x,\text{NWT}}, \tag{10}$$

where $r_{\text{WT}}$, $r_{\text{PLD}}$ and $r_{\text{GT}}$ denote the ratio of bare soil cells in the land cover matrix covered by water tracks, paleolake deltas and glacial till, respectively, and $Q_{\text{WT}}$ and $Q_{\text{NWT}}$ signify energy fluxes at WT and NWT, respectively.

## 3   Results and discussion

Differences in surface energy fluxes between PLD and GT were $42 \pm 31$ % smaller compared to those between WT and NWT, judging from root mean-squared errors (see Appendix B). We hence present the results by comparing mean observations between WT and NWT – the latter being ensemble-averaged values over PLD and GT – assuming that the recording period was sufficiently long to calculate meaningful average energy fluxes.

The summer in this high-latitude MDV ecosystem is unusually short. The observations over 26 days were taken during the peak summer season centered around the end of December because of maximum insolation. If defining summer as the period during which the ice contained in the ground at some depth melts as indicated by temperatures equal to or above freezing, then

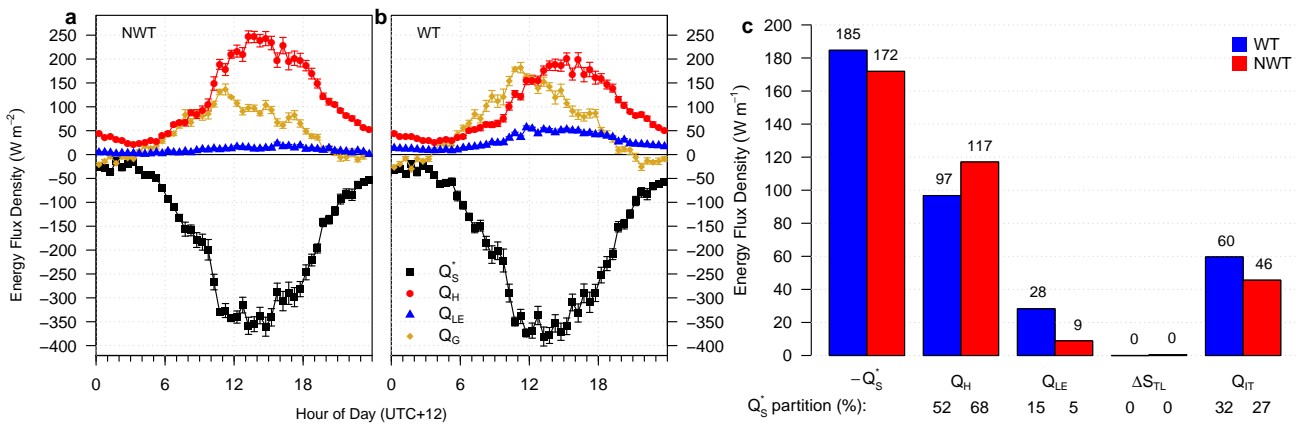

**Figure 4.** Surface energy balances for NWT (non-water track) and WT (water track) averaged over the whole measuring period, including net radiation ($Q_S^*$), sensible heat flux ($Q_H$), latent evaporative heat flux ($Q_{LE}$) and soil heat fluxes. **(a-b)** Ensemble mean diurnal variations including ground heat flux ($Q_G$); negative fluxes are directed to the ground surface, positive fluxes away from it; error bars depict standard errors of the mean. **(c)** Ensemble averages including energy storage change in the thawed layer ($\Delta S_{TL}$) and soil heat flux at ice table depth ($Q_{IT}$); partitions of $Q_S^*$ at WT and NWT are added in % for each heat flux.

the length of the summer equates to 95, 82, or 67 days when considering the long-term observations from the MDV Long Term Ecological Research (LTER) site over the period 1993 to 2011 at depths 0, 0.05, or 0.10 m, respectively. The observational period therefore represents a substantial portion of the MDV summer during peak season.

Averaged over the entire observational period of 26 days, more than 50 % of the net radiation $Q_S^*$ was transferred to sensible
5   heat flux $Q_H$ at both NWT and WT, $\approx$30 % was taken up by frozen soil beneath the ice table ($Q_{IT}$), and the remainder was transferred to latent evaporative heat flux $Q_{LE}$. The net energy flux in the thawed layer ($\Delta S_{TL}$) was insignificant leading to $Q_{IT} \approx Q_G$ (Fig. 4). Similar partitioning of $Q_S^*$ was found at high-arctic tundra heath in Greenland and polar semi-desert in Svalbard (Lloyd et al., 2001; Lund et al., 2014), though $Q_{LE}$ of NWT contributed much less to the SEB than in moister arctic sites, which emphasizes the extremely dry conditions in the MDV.
10   Since meteorological forcings were nearly indentical at WT and NWT (see supplementary Fig. 3) lending to their close proximity, we argue that differences in SEB components are caused by the variation in surface properties and soil water content.

While our measurements show conditions of peak summer warming, we assume that for the remaining part of the year a radiative net loss occurs on water tracks and non-water tracks due to lack of insolation. We also expect water tracks to lose more
15   heat during the winter because of higher thermal diffusivity in the soil, and to show increased sublimation owing to the high ground ice content. For quantification of energy exchange of water tracks and non-water tracks on an annual scale year-round measurements are needed.

## 3.1 Net radiation

Magnitudes and average fluxes of $Q_S^*$ were increased by a factor of 1.1 at WT relative to NWT (Fig. 4) determined from a linear correlation ($R^2 = 0.99$, $p < 0.001$). A lower albedo was observed at WT ($0.13 \pm 0.01$) compared to NWT ($0.16 \pm 0.01$), averaged over radiation measurements between 11:00 and 13:00 UTC+12. We interpret that the increased $Q_S^*$ at WT were
partly caused by the reduced albedo (Levy et al., 2011). Levy et al. (2014) found even greater differences in albedo 0.22 for on-track and 0.14 for off-track soils, suggesting that the difference in energy uptake between water tracks and dry soils may even be greater at other locations.

    Lower radiative surface temperatures were observed for WT ($5.0 \pm 3.2$ °C) than for NWT ($6.3 \pm 5.1$ °C) when averaging over the entire record. Differences in surface temperatures between the stations were lowest at low solar angles and greatest
at high solar angles, were temperatures reached up to 12.7 °C for WT and 19.5 °C for NWT. These observations suggest that increased $Q_S^*$ at WT relative to NWT can be partly explained by the reduced energy loss through upwelling longwave radiation due to lower surface temperatures which are caused both by higher evaporative cooling and higher soil heat capacity (Tab. 2). This reduced surface heating at WT explains why the differences in $Q_S^*$ between WT and NWT are greatest around solar noon when solar zenith angles are smallest (Fig. 5).

## 3.2 Turbulent heat fluxes

Averages of the combined turbulent heat fluxes $Q_H + Q_{LE}$ reached $\approx 130$ W m$^{-2}$ at both WT and NWT (Fig. 4c). The partitioning between $Q_H$ and $Q_{LE}$, however, strongly differed: At WT magnitudes of $Q_{LE}$ computed from linear correlation ($R^2 = 0.84$, $p < 0.001$) were increased by a factor of 3.0 relative to NWT, while magnitudes of $Q_H$ were reduced to 0.7 ($R^2 = 0.97$, $p < 0.001$). Average heat fluxes at WT were increased by a factor of 3.2 for $Q_{LE}$ and reduced to 0.8 for $Q_H$
compared to NWT (Fig. 4c). Mean Bowen ratio $Bo = Q_H \, Q_{LE}^{-1}$ over the measuring period was 3.3 for WT and 15.5 for NWT. $Q_{LE}$ peaked about 1 hour earlier during the day and $Q_H$ 1 hour later at WT compared to NWT. Turbulent heat fluxes reached their daily maximum in the afternoon irrespective of the surface (Fig. 5).

    Increased water content of the water track soil (Langford et al., 2015; Levy et al., 2011, 2014) was likely the cause for increased $Q_{LE}$ and reduced $Q_H$, confirming the findings in high arctic environments (e.g. Westermann et al., 2009). The high
soil moisture also led to the earlier, longer and more pronounced daily peak of $Q_{LE}$ at WT due to the greater heat diffusivity.

**Table 2.** Measurements of ice table depth as depth-to-refusal DTR (m) and soil thermal properties close to the surface: soil volumetric heat capacity $C_G$ (MJ m$^{-3}$ K$^{-1}$), thermal conductivity $K_G$ (W m$^{-1}$ K$^{-1}$), thermal diffusivity $D_G$ (mm$^2$ s$^{-1}$); mean $\pm$ standard deviation). Measurements were taken for WT (water track) and PLD (paleolake delta).

|  | DTR | $C_G$ | $K_G$ | $D_G$ |
|---|---|---|---|---|
| WT | 0.48 | 2.29±0.25 | 1.19±0.04 | 0.52±0.05 |
| PLD | 0.30 | 1.32±0.12 | 0.39±0.17 | 0.29±0.12 |

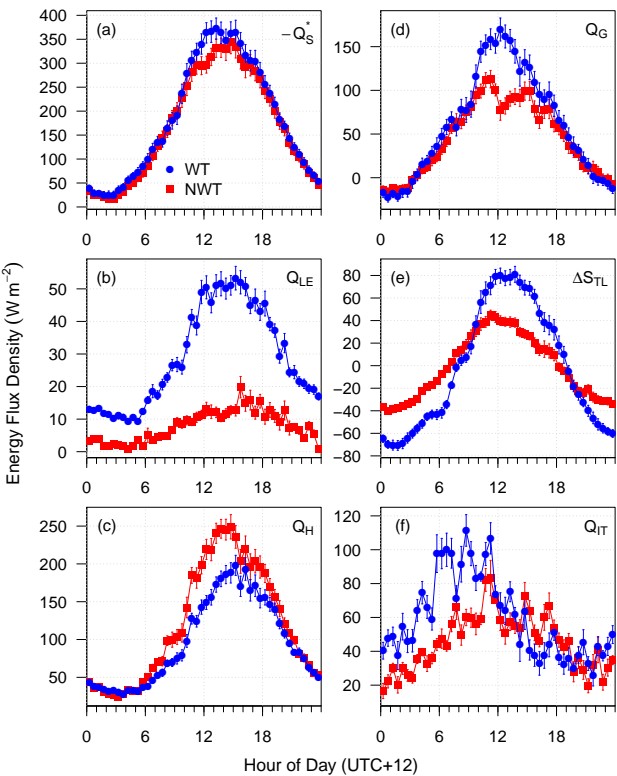

**Figure 5.** Comparison of diurnal variations of inverted net radiation $-Q_S^*$ **(a)**, latent evaporative heat flux $Q_{LE}$ **(b)**, sensible heat flux $Q_H$ **(c)**, ground heat flux $Q_G$ **(d)**, energy storage change in the thawed layer $\Delta S_{TL}$ **(e)** and soil heat flux at ice table depth $Q_{IT}$ **(f)** between WT (water track) and NWT (non-water track), averaged over the entire measuring period of 26 days. Energy fluxes for **(a)** and negative energy fluxes for **(d)** are directed towards the ground surface, energy fluxes for **(b)-(d)** and **(f)** and positive energy fluxes for **(d)** point away from the ground surface. Positive energy fluxes for **(e)** are directed from the ground surface or the ice table into the thawed layer and negative fluxes move out of the thawed layer towards the ground surface or the ice table. Error bars depict standard errors of the mean.

The resulting strong evaporative cooling constrained $Q_H$ by weakening surface warming at WT. This finding likely explains why $Q_H$ showed a later and less distinct daily maximum at WT compared to NWT, where energy is preferentially exchanged through $Q_H$. The maximum turbulence intensity was observed in the late afternoon hours, which led to the concurrent peak in turbulent fluxes.

### 3.3 Conductive heat fluxes

Mean soil temperatures in the topmost 0.3 m of the soil were lower for WT (2.9 ± 2.0 °C) than for NWT (3.7 ± 2.6 °C) (see also supplementary Fig. 4), contradicting previous observations that soil temperatures of water tracks are higher than their surroundings in summer (Levy et al., 2014). The depth-to-refusal measurements yielded greater ice table depths for WT than for PLD (Tab. 2), which confirmed contrasting active layer depths between water tracks (0.45 m) and dry adjacent soils

(0.19 m) observed in Lake Hoare and Lake Bonney basins (Levy et al., 2011; Gooseff et al., 2013). Modeled active layer depths for lower Taylor Valley of 0.45 to 0.75 m (Bockheim et al., 2007) indicate that in our experiment the active layer was not fully thawed and the ice table was still being lowered.

Linear correlation analysis of the conductive heat fluxes between WT and NWT yielded an increase by a factor of 1.4 for $Q_G$ ($R^2$ = 0.93, p < 0.001) and by 1.8 for $\Delta S_{TL}$ ($R^2$ = 0.92, p < 0.001) at WT. Average diurnal variations of $Q_{IT}$ showed a substantial scatter preventing a meaningful correlation. Average heat fluxes at WT were increased by a factor of 1.3 for $Q_G$ and $Q_{IT}$ relative to NWT while they were zero for $\Delta S_{TL}$ (Fig. 4). The distinctly higher $\Delta S_{TL}$ at WT compared to NWT at the time of the daily maximum can be explained by the higher thermal diffusivity of the water track soil compared to dry paleolake delta sediments (Tab. 2) due to increased soil moisture mentioned earlier (Ikard et al., 2009; Levy et al., 2011).

On average, $Q_G$ was directed into the soil except for the period of low solar elevation angles between 21:30 and 03:30 UTC+12 when the net energy transported was directed out of the ground to the surface. $\Delta S_{TL}$ changed sign approximately every 12 hours, turning the thawed layer from an energy sink into a source and back. $Q_{IT}$ remained positive throughout the day when averaged over the entire observational period. The simultaneous occurrence of negative $Q_G$ and positive $Q_{IT}$ at low solar elevation angles reveals a flux divergence in $\Delta S_{TL}$ which was directed both towards the ground surface and the ice table at these times, leading to significant net energy loss from the buffering thawed layer.

Since the active layer is typically not fully thawed before mid-January in the MDV (Adlam et al., 2010; Conovitz et al., 2006), we argue that the ice table was being lowered throughout the recording period and most of $Q_{IT}$ was consumed by latent heat of thawing the frozen layer which explains the large energy fluxes matching soil heat flux observations in other permafrost regions (Lloyd et al., 2001; Lund et al., 2014; Westermann et al., 2009). The increased $Q_{IT}$ at WT relative to NWT was likely due to larger amount of ice in the frozen layer and stronger transport of energy through the thermally more conductive thawed layer.

The mean diurnal peak in $Q_{IT}$ occurred 2.5 hours earlier for WT compared to NWT (Fig. 5). Calculation of the diurnal temperature wave penetration lag $t_{\text{lag}}$ from the surface to ice table depth with Eq. (9) yielded a 15 and 13 hours lag for WT and NWT, respectively. This finding suggests that the observed daily peak in $Q_{IT}$ at one specific day corresponded to the peak in $Q_G$ from the previous day delayed by 21 and 24 hours for WT and NWT, respectively. The relatively earlier peak of $Q_{IT}$ at WT lends strong support to our claim of a faster transport of energy through the thawed layer due to its increased thermal diffusivity in water tracks (Tab. 2) despite the greater depth of the thawed layer.

As mentioned above, our findings of $Q_{IT}$ need to be carefully discussed in the context of potential artifacts arising from a non-zero observational residual in the SEB, which could obscure true physical behavior of this significant SEB term in permafrost-dominated ecosystems. On one hand, the increased scatter in the correlation analysis between WT and NWT points to some random error in $Q_{IT}$ estimated as the residual of the SEB. Random error is largest for the residual term as the random uncertainty of all other directly measured or observation-based computed terms is projected into variations of $Q_{IT}$, while the individual random errors may not compensate each other for an individual averaging period and across the observational period.

On the other hand, many findings support the claim that our estimates of $Q_{IT}$ represent the true physical heat transport and ice melting process: (i) $Q_{IT}$ was increased by 30 % at WT compared to NWT, which over the course of the summer melting season should lead to deeper thawed layer and lower ice table depth both confirmed by our DTR measurements. (ii) The sign of $Q_{IT}$ was always positive and its magnitude was similar to that of $(Q_H + Q_{LE})/2$ confirming that a significant fraction of net radiation is consumed for ice table lowering by melting in the peak summer season. The magnitude of our $Q_{IT}$ estimates were similar to those found in other permafrost landscapes. If $Q_{IT}$ were dominated by $\epsilon$, then the residual would correspond to roughly 30 % of $Q_S^*$, which is the maximum observed value for non-permafrost surfaces. Given the relative homogeneity in surface conditions in MDV compared to that in vegetated non-permafrost surfaces, it is unsure whether such a large residual would be realistic. (iii) The diurnal course of $Q_{IT}$ showing an earlier peak for WT matched our expectation from the time lag analysis in combination with the soil thermal property measurements. Those would lead to a small dominance of the effect of enhanced thermal diffusivity over the greater transport time required for the daily heat pulse from the ground surface to penetrate to the ice table depth due to the greater distance to be covered. One must recall that the thermal diffusivity is the ratio of the energy conducted to that stored in a given medium. (iv) The oscillatory nature of the measured $\Delta S_{TL}$ over the course of the day equating to zero when averaged over the ensemble-averages at any location lends strong support to our interpretation as the temporary heat storage reservoir and thus heat communicational layer. We claim that when it is combined with the other measured SEB terms in the form of $-Q_S^* - Q_H - Q_{LE} - \Delta S_{TL}$ it forces physically meaningful diurnal dynamics in $Q_G$ and $Q_{IT}$.

However, the existence of some influence of a true SEB observational residual $\epsilon$ was confirmed, but judging from our observations it induces a random component in the diurnal course of the ensemble-averaged components and is not expected to have a systematic diurnal signal leading to an unphysical timing or magnitude in $Q_{IT}$. Our method is unable to rule out the existence of a constant, time-invariant systematic $\epsilon$ which would effect changes in the magnitude of $Q_{IT}$. However, we would expect its magnitude and thus its impact to be small. An independent verification of our method to estimate $Q_{IT}$ as the SEB residual could have been gleaned from observations extending into the after-summer season when the ice table depth is decreasing due to the refreezing of the soil moisture. Under those conditions, a change in sign of $Q_{IT}$ would be expected which could lend support to a physically meaningful quantification of all SEB terms. Unfortunately, we were unable to extend the observational period for logistical reasons.

### 3.4 Climate change response of soils

How will expected 21st century warming in the MDV (Arblaster and Meehl, 2006; Chapman and Walsh, 2007) impact mass and energy fluxes of soils? Though water tracks and other wetted soils show high spatial consistency in the MDV, their spatial extent is largely dependent on snow and ground ice melt (Gooseff et al., 2013; Langford et al., 2015; Levy, 2015).

Snow abundance, distribution, and duration will be an important factor for predicting changes to wetted area, since snow is a major water source for many water tracks in the MDV (Langford et al., 2015). Snowfall is projected to rise during the next century (Christensen et al., 2013), and snowdrift accumulation, providing an amount of snow equal to snowfall in the valley floors, will mainly depend on dynamics in katabatic winds delivering snow from the valley walls and ice sheet (Fountain et al.,

**Table 3.** Integrated energy fluxes from bare soil in lower Taylor Valley calculated with Eq. (10). "Factor" denotes different increments of water track coverage WTC relative to the observed value of 2.9 %. Net radiation ($Q_S^*$), sensible heat flux ($Q_H$), latent evaporative heat flux ($Q_{LE}$) and ground heat flux ($Q_G$) in $\mathrm{W\,m^{-2}}$.

| Factor | WTC (%) | $-Q_S^*$ | $Q_H$ | $Q_{LE}$ | $Q_G$ |
|--------|---------|----------|-------|----------|-------|
| 1.0 | 2.9 | 172.3 | 116.5 | 9.4 | 46.4 |
| 1.5 | 4.4 | 172.5 | 116.2 | 9.7 | 46.6 |
| 2.0 | 5.9 | 172.7 | 115.9 | 10.0 | 46.8 |
| 3.0 | 8.8 | 173.0 | 115.3 | 10.6 | 47.2 |

2009). Melt of existing snow patches in the summer is likely to increase in response to a warming climate. As water tracks also feed on ground ice melt (Harris et al., 2007), deepening active layer thaw is all but guaranteed to produce additional melt. This, in turn, may drive subsidence, which could trap new snow banks, encouraging further water track activity. Therefore, we expect water tracks to expand in surface fraction cover and activity in response to climate change: (i) as active layers thaw more deeply and melt previously-stable ground ice due to positive feedbacks of increased thermal diffusivity and energy uptake of wet soils (Gooseff et al., 2013; Ikard et al., 2009; Levy and Schmidt, 2016), (ii) as perennial snow patches melt and thin during summer warming, and (iii) as increased evaporation from the Ross Sea and decreased sea ice raises humidity, allowing enhanced soil salt hydration. All these mechanisms collectively or individually point to the potential for expanded surface fractions and connectivity of wetted soil in the MDV (Ball et al., 2011; Wall, 2007). This anticipated development of water track abundance is contrary to projections for the High Arctic where primarily snow-fed water tracks are likely to be reduced in area because precipitation will increasingly occur in liquid form (Comte et al., 2018).

To provide some quantification of the anticipated response of energy and matter fluxes in the MDV in the summer season based upon our evaluation of the SEB of water tracks, we calculated the increase in average energy fluxes integrated for bare soil in the entire lower Taylor Valley resulting from arbitrary, but realistic increases in the relative water track fraction of bare soil using Eq. (10) (Tab. 3). A doubling scenario is a realistic increase in water track coverage considering that the spatial extent of wetted soils in the MDV varies by at least a factor of 2 between cold and warm years (Langford et al., 2015). This doubling would lead to an increase in evaporation from bare soil in the entire lower Taylor Valley by 6 % to $0.36\,\mathrm{mm\,d^{-1}}$, which implies a significant increase of evaporation in the MDV given the expected response mechanisms in soils mentioned above.

One likely implication of our results demonstrating the intensified heat cycling for water tracks is that internal physical processes of water tracks, i.e., energy exchange and hydrology, could respond faster and more vigorously to climate change than dry off-track soils in addition to an increase in coverage in the MDV. Since increased thermal conductivity and energy uptake enhance positive soil thawing feedbacks for water tracks (Gooseff et al., 2013; Ikard et al., 2009; Levy and Schmidt, 2016), soil moisture may rise and active layers deepen, leading to increases in discharge and evaporation compared to the current conditions. These effects can be assessed in future monitoring efforts for water tracks, where observed trends in internal processes of water tracks would serve as indicators for landscape-scale climate change response in the MDV.

## 3.5 Climate change scenarios

Since we have presented many indications of increasing spatial extent of water tracks with climate change in the MDV, a discussion of possible pathways of climate change and their respective roles in determining the change in spatiotemporal dynamics of water tracks will benefit the framing of energy and mass flux scenarios for the MDV, and could serve as a basis for advanced micro-scale modeling to provide constraints on the timing and spatial feedbacks of these predicted mechanisms. Here we briefly consider two contrasting scenarios informed by meteorological first principles: One where insolation increases but regional air temperatures remain constant (Gooseff et al., 2017), and one where temperatures increase but insolation remains constant, which corresponds to a more typical Arctic-type climatology.

While we lack sufficient information to make quantitative estimates for the two alternative scenarios, we can offer predictions based upon mechanistic reasoning: In the first scenario of constant insolation and rising temperatures in the MDV, an increase in vapor pressure deficit would lead to delayed cloud formation while net radiation would increase because of reduced cloud coverage, causing a positive feedback on temperature. The increased temperature would act to intensify snowmelt and soil thawing. This effect would be even enhanced by the occurrence of strong down-valley winds which can increase temperatures and melting rates in the summer dramatically (Doran et al., 2008). In the second, alternate scenario temperatures stay constant and insolation increases as observed by Gooseff et al. (2017). Higher net radiation will cause an evaporation increase which will reduce the vapor pressure deficit since air temperatures remain constant. The increased cloud formation would then cause a negative feedback on insolation and lead to a weaker increase in snow and ground ice melting than in the first scenario. Both scenarios represent pathways of climate change which would increase spatial extent of water tracks and other wetted soils, though the magnitude of this climate change response may vary between them.

## 3.6 Implications for soil ecosystems

Here, we speculate on the implications of the differences in SEB found between water tracks and dry soils onto the biologic communities and geochemistry: Since water tracks are wetter than adjacent off-track soils at the surface, it is possible that these conditions promote enhanced chemical weathering of the water track soils (Campbell et al., 1998). One might expect that with expanded water tracks, solute fluxes to water tracks would increase as chemical weathering initiates on what were cold, dry soils. Increased weathering could also result in increased Phosphorous fluxes to soils, enhancing microbial and invertebrate habitat suitability (Heindel et al., 2017). Our SEB findings suggest that water tracks go through fewer freeze-thaw cycles than adjacent off-track soils, due to the release of latent heat during freezing. This reduces biotic stresses, which in turn might improve habitat suitability for both microbes and invertebrates (Yergeau and Kowalchuk, 2008; Convey, 1996). Active layer depths are typically several dm deeper in water tracks than they are in adjacent dry soils (Levy et al., 2011). That suggests that occupation of currently dry soils by water tracks may result in an expansion of the seasonally thawed portion of the soil column, producing expanded habitat for soil invertebrates.

However, these effects may only lead to increase in biomass, productivity and respiration if episodic overland flow flushes solutes out of the wetted soil (Ball and Levy, 2015; Zeglin et al., 2009). Overland flow can only occur when meltwater availabil-

ity exceeds infiltration capacity and hydrologic conductivity of water track soils (Levy, 2015). In combination, the discussed processes could lead to more hydrologically connected and less saline soil habitats at increased and more uniform soil moisture in the MDV and, by that, increased biogeochemical cycling (Ball et al., 2011; Gooseff et al., 2013). This habitat change could suppress dry-tolerant taxa like the dominant nematode *Scottnema lindsayae* and favor wet-tolerant generalist taxa at lower

biodiversity levels (Buelow et al., 2016; Simmons et al., 2009; Wall, 2007). However, it is likely that overall improved habitat suitability would increase biological productivity and respiration in soils in the MDV.

## 4  Conclusions

This study aimed at characterizing the energy exchange in the summer season across contrasting surfaces of water tracks and non-water tracks in lower Taylor Valley, McMurdo Dry Valleys, Antarctica. Non-water tracks are composed of glacial till and

paleolake delta sediments. Our findings support our initial hypothesis that water tracks significantly modify the surface energy budget of the terrestrial Antarctic due to lowered surface albedo and increased water content in the active layer, leading to an enhanced cycling of heat, water, and potentially carbon. Our measurements are representative for summer conditions, since they were taken during peak summer warming and cover at least one third of the summer season.

Water tracks showed enhanced net radiation, latent evaporative heat flux, and soil heat flux in the thawed and frozen layers,

as well as smaller sensible heat flux relative to the reference off-track locations. For both water tracks and non-water tracks, the sensible heat flux was the largest SEB term consuming $\approx 50\,\%$ of the net radiation. The second largest energy sink was the soil heat flux at ice table depth, which corresponded to $\approx 30\,\%$ of the net radiation and mainly provides the melting energy lowering the ice table as summer season progresses. Its magnitude was larger than that of the latent evaporative heat flux. Considering a realistic scenario of an increase in abundance of water tracks with the observed properties in lower Taylor Valley would cause

a considerable rise in evaporation from this Antarctic landscape.

In summary, our findings provide convincing evidence that water tracks in the McMurdo Dry Valleys have a strong impact on the surface energy balance in the summer season, particularly on the latent evaporative heat flux, by increasing the uptake of energy at the surface. Our study isolates the physical effect of soil moisture on energy and matter exchange in cold ecosystems in the absence of vegetation. Considering the anticipated increase in area covered by water tracks and other wetted soils in

the MDV in response to a warming climate, our results point to an increasing importance of water tracks as features in the hydrological system of polar deserts which will depend on the exact forcing mechanism either by changes in temperature or insolation. The expected expansion of water track coverage in MDV soils could provide access to new soil habitats with altered habitat suitability, affecting biological activity across the MDV.

## Appendix A:  Sensitivity of energy storage change in the thawed layer to ice table depth

Since only one measurement of ice table depth $z_{IT}$ for WT and NWT was taken each, the sensitivity of $\Delta S_{TL}$ to variations in $z_{IT}$ was analyzed. We assumed that the thawed layer was spatially heterogeneous and a temporal increase in $z_{IT}$ owing to

**Table A1.** Using root mean-squared errors (RMSE) calculated with Eq. (A1) for the comparison of mean diurnal variations of energy storage change in the thawed layer ($\Delta S_{TL}$) over the whole measuring period between parameters $x_1$ vs. $x_2$. **(a)** Comparison between WT (water track) and NWT (non-water track) values with measured ice table depths ($z_{IT}$). **(b)** Sensitivity of $\Delta S_{TL}$ to different $z_{IT}$ values at WT, where measured $z_{IT}$ is assigned to $x_2$. **(c)** Same as **(b)**, but for NWT.

| | **(a)** WT vs. NWT | **(b)** WT Sensitivity | **(c)** NWT Sensitivity |
|---|---|---|---|
| $x_1$ | $\Delta S_{TL,WT}(z_{IT} = 0.48\ m)$ | $\Delta S_{TL,WT}(z_{IT} = 0.30\ m)$ | $\Delta S_{TL,NWT}(z_{IT} = 0.48\ m)$ |
| $x_2$ | $\Delta S_{TL,NWT}(z_{IT} = 0.30\ m)$ | $\Delta S_{TL,WT}(z_{IT} = 0.48\ m)$ | $\Delta S_{TL,NWT}(z_{IT} = 0.30\ m)$ |
| RMSE ($W\ m^{-2}$) | 27.62 | 7.18 | 3.90 |

the thawing process was occurring. To quantify this sensitivity, we compared $\Delta S_{TL}$ calculated using measured $z_{IT} = 0.48$ m at WT and $z_{IT} = 0.30$ m at NWT with alternative $\Delta S_{TL}$ resulting from arbitrarily assuming $z_{IT} = 0.30$ m at WT and $z_{IT} = 0.48$ m at NWT. The root mean-squared errors (RMSE) were computed as

$$\text{RMSE} = \sqrt{\frac{1}{48} \sum_{i=1}^{48} (x_1 - x_2)^2}, \tag{A1}$$

where $i$ runs from 1 to 48 to include all half-hour values of the mean diurnal variations and $x_1$ and $x_2$ correspond to $\Delta S_{TL}$ values specified for three cases (Tab. A1): (a) $\Delta S_{TL}$ with measured $z_{IT}$ was compared between WT and NWT. (b–c) RMSE between $\Delta S_{TL}$ computed using measured vs. alternative $z_{IT}$ was determined for WT and NWT, respectively, quantifying the sensitivity of $\Delta S_{TL}$ to uncertainty of $z_{IT}$ for both WT and NWT.

    This senstivity of $\Delta S_{TL}$ to variation of $z_{IT}$ expressed by RMSE was 15 % and 16 % of average $\Delta S_{TL}$ magnitudes for WT
and NWT, respectively. RMSE between WT and NWT was around 4 and 7 times as high as the RMSE sensitivity estimates for WT and NWT, respectively (Tab. A1), which provides evidence that sensitivity of $\Delta S_{TL}$ to uncertainty of $z_{IT}$ is negligible compared to the impact of the observed water track on $\Delta S_{TL}$ relative to non-water tracks.

## Appendix B:  Variations in surface energy balance components across different reference locations

    PLD was located in a paleolake delta and was characterized by fine material, while the dominant glacial till at GT had a coarse
texture. Differences in texture and other properties of the surfaces, e.g. albedo, may lead to considerable differences in surface energy fluxes between PLD and GT surfaces. These disparities between PLD and GT were compared to differences between WT and NWT via two statistical estimates, RMSE1 and RMSE2, to examine if the influence of the water track on the SEB is more significant than the effect of other surface properties.

    $\text{RMSE1}_x(t_j)$ represents the differences in mean diurnal variations of energy fluxes between WT and NWT for each SEB
component $x$, where $t_j$ specifies the periods $t_1$ and $t_2$, i.e., the recording periods of PLD and GT, respectively. $\text{RMSE1}_x(t_j)$ was calculated using Eq. (A1) with $x_1 = Q_{x,WT,i}(t_j)$ and $x_2 = Q_{x,NWT,i}(t_j)$.

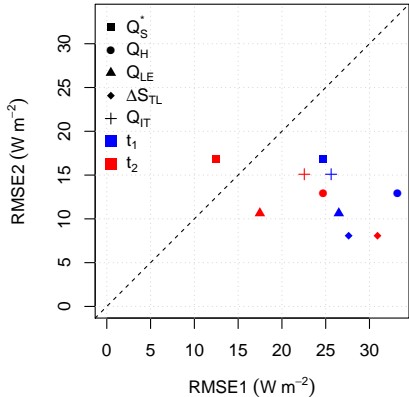

**Figure B1.** Investigating differences between surface energy fluxes at PLD (paleolake delta) and GT (glacial till), both belonging to NWT (non-water track reference stations), via comparison of two statistical estimates using mean diurnal variations and based on root mean-squared errors (RMSE): RMSE1 quantifies differences between WT (water track) and NWT separately for recording periods of PLD ($t_1$) and GT ($t_2$). RMSE2 computes disparity of PLD and GT, cleared of the influence of meteorological boundary conditions, calculated as the change from $t_1$ to $t_2$ of differences between WT and NWT. Computation of RMSE1 and RMSE2 for net radiation ($Q_S^*$), sensible heat flux ($Q_H$), latent evaporative heat flux ($Q_{LE}$), heat storage change in the thawed layer ($\Delta S_{TL}$) and soil heat flux at ice table depth ($Q_{IT}$).

RMSE2$_x$ compares differences between surface energy fluxes of WT and NWT stations at $t_1$ to those at $t_2$. Equation (A1) was applied for calculating RMSE2$_x$ with $x_1 = Q_{x,WT,i}(t_1) - Q_{x,NWT,i}(t_1)$ and $x_2 = Q_{x,WT,i}(t_2) - Q_{x,NWT,i}(t_2)$. Thus, RMSE2$_x$ represents the disparity of PLD and GT, compensating for temporal changes of absolute energy flux values caused by varying meteorological conditions between $t_1$ and $t_2$.

For most SEB components, RMSE1$_x(t_j)$ exceeded RMSE2$_x$. Only $Q_S^*$, RMSE1$_{Q_S^*}(t_2)$ was smaller than RMSE2$_{Q_S^*}$ (Fig. B1). On average, RMSE2$_x$ was $42 \pm 31$ % smaller than the corresponding values of RMSE1$_x(t_j)$. This indicates that for most SEB components and both PLD and GT, the differences between water track and reference surfaces were larger than between the two reference surfaces, except for $Q_S^*$ at GT. Hence, the impact of water tracks on surface energy fluxes was generally more important than the effect of soil texture and other properties varying between PLD and GT. For $Q_S^*$, differences between GT and WT were less significant than the difference between PLD and GT, which can be explained by very similar albedo at GT and WT.

## Appendix C: Soil thermometers

This section contains information about dates and locations of soil temperature measurements.

**Table C1.** Soil thermometer devices, depths of their deployment and recording periods for the stations WT (water track), PLD (paleolake delta) and GT (glacial till).

| Station | TC Depths (cm) | TC Recording Periods | TR Depths (cm) | TR Recording Periods |
|---|---|---|---|---|
| WT | 2×0.4 | 26 December 2012–04 January 2013 | 1,4,12,22 | 04 January 2013–21 January 2013 |
| | 0.4 | 04 January 2013–21 January 2013 | | |
| PLD | 2×0.4 | 27 December 2012–04 January 2013 | 4,7,30 | 28 December 2012–14 January 2013 |
| | 1,4,12 | 04 January 2013–14 January 2013 | | |
| GT | 1,4,12 | 14 January 2013–21 January 2013 | - | - |

TC: Thermocouples (*TMTSS-020, OMEGA Engineering Inc., Norwalk, CT, USA*).

TR: Thermistors, recorded with *HOBO H8 Pro* logger from *Onset Computer Corp., Bourne, MA, USA*.

*Author contributions.* CT designed the experiment. JL and CT carried it out, where JL focused on soil measurements and CT was responsible for eddy-covariance and radiation measurements. TL performed the footprint simulations and analyzed the results with support from CT. TL prepared the manuscript with contributions from all co-authors.

*Competing interests.* The authors declare that they have no conflict of interest.

5 *Acknowledgements.* This publication was funded by the German Research Foundation (DFG) and the University of Bayreuth in the funding programme Open Access Publishing. This work was supported in part through NSF OPP award # 1343649 to Joseph S. Levy. We thank Sherri Johnson, US Forest Service, for making the net radiometers available for the field campaign. Footprint modeling with TERRAFEX was assisted by Wolfgang Babel from BayCEER.

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
