# Peer review of "Water tracks intensify surface energy and mass exchange in the Antarctic McMurdo Dry Valleys"

_The Cryosphere, 2019_

## Referee Comment (RC1) · Anonymous Referee #1 · 12 Feb 2019

General comments The authors use eddy-covariance to compare late summer surface energy balance between a water track and two non-water track (reference) sites in Taylor Valley, Antarctica. They demonstrate that the water track site registers greater energy exchanges and greater relative contributions of evaporative and permafrost heat fluxes to the surface energy balance, while registering lower sensible heat fluxes. It provides a rare account of the effect of wetness on soil thermal behavior, with much implications for biogeochemical and hydrological processes, and is highly relevant in order to further identify the implications of climate change for periglacial and polar desert soils. The absence of vegetation on the site draws on the importance of research on naked watersheds in order to isolate the effect of soil hydrology on its physical, chemi-

cal and thermal properties and behavior. The overall originality and presentation quality of the manuscript is good, clearly providing the readers with greater understanding of energetic processes operating in water tracks and in polar desert soils in general. At this point, however, the manuscript only presents fair scientific qualities and significance. This is not a criticism of the method and results, but a consequence of the overall theoretical framework of the manuscript and of the lack of discussion and perspective on the work. While the results are sound, the manuscript would benefit from modifications in order to correctly present the narrative behind the measurements, to better address the significance of the findings and to put them in greater perspective and in accord with the existing literature. The previous statements could be framed within two main criticisms. The first concerns the ergodic theoretical framework used to compare between water track and reference SEB, in turn rooted in the climate change paradigm used by the authors to present their work. In order to substitute space to time, the authors assume the landscape will become "wetter" with new water tracks appearing in front of (new?) snowdrift sites or downslope near the valley bottom. It is therefore central to the premise of the manuscript to clearly demonstrate how climate change will increase the spatial distribution of water tracks, using the literature and possibly a conceptual model, yet it is not established clearly enough in the introduction. Specifically, there are some questions that arise on the mechanisms by which dry areas should become water tracks (see specific comments), which goes against earlier findings (see Langdon et al. 2014). In summary, change is not what is measured here, and better definitions and demonstrations of "change" need to be included in order to anchor the scientific claims of the paper to reality. The second criticism concerns the lack of perspective in the results and discussion section, and to some degree in the introduction. In section 5, a single reference to the literature can be found outside of the first paragraph of section 5.4, and nothing is done to link the findings to other similar comparisons done in the area and to general hydrological, biogeochemical and ecosystemic research concerning the MDV and water tracks, some which were already cited (Ball et al. 2011; Ball & Levy 2015; Ikard et al. 2009; Fountain et al. 2014; Levy et

al. 2013; 2016; Schmidt and Levy, 2017; Gooseff et al. 2011; Steven et al. 2013; Paquette et al. 2017; 2018; Comte et al. 2018; Zeglin et al. 2009; 2011; also, see water track literature from Alaska). It is essential to root the research into the existing literature, and to relate the findings to what has already been observed. As the manuscript appears now, it does not demonstrate a thorough understanding of the literature on the subject. In addition, further attention to the specific nomenclature of permafrost soils is required.

Specific comments p.1 line 7: Sentence needs to be clarified Line 9: 30% to melting the seasonally thawed layer : The active layer is already thawed at the onset of experimentation, so much so that it is considered of a stable depth in the calculations. How is it then that 30 % of the heat is transferred to melting it? Is it meant as warming permafrost under what is called the ice table (QIT)? Line 13-14: The evaporation from lower TV considers land only or also water bodies (Lake Fryxell, rivers)? Line 15: This is a bit overstated, as the manuscript does not address the effect of adding or removing water tracks. Also, ice-sheet free Antarctic regions could be changed to "polar deserts" to broaden perspectives. Line 16: . . . are likely to respond faster to climate change signals. . . How are they going to respond? Their hydrology is going to change? Their SEB will change? This is never addressed nor measured in the manuscript, the only landscape change premised is the passage from dry to wet soils, which appears as the main signal of climate sensitivity on the slope. Therefore, if slopes become wet and water track occurrence increases, then the water tracks are rather resilient to change, and their distribution will even "benefit" from climate change. Also, are the potential changes responses to climate change signals or to climate change? In addition, Langdon et al. (2014) have shown that climate change may cause increases in water track activity, but that they show spatial consistency in their location, since they highly depend on snowdrift accumulation. Line 17-18: Their spatiotemporal dynamic will be an effect of climate change, but not of sensitivity to it, unless reference sites are discussed here. Line 25: well-documented: Citations needed. p.2 Line 3: could use citation from Gooseff et al. 2016 Line 7: This definition is very regional to TV. A better definition for

water tracks can be found in Gooseff et al. 2013. Line 8-10: This statement fails to explain why water tracks are more sensitive to climate change than non-water tracks. Line 16-17: This sentence is the prime assumption to the general "change in the face of climate change" message of the manuscript. It is however not well documented and demands to be proven before the "change" paradigm can be accepted. Line 17: Here we identify an opportunity to investigate the utility of this potentially useful indicator... This is never really what this research is about, as the utility of the indicator (are water tracks really indicators?) is not investigated. Line 21: Latent heat flux is used throughout the manuscript to refer to latent evaporative flux. Since this is a permafrost area and two changes of state are possible, it would be suitable to include evaporation in the wording. Line 30: ice table. Please explicitly define this term. It appears to designate the thaw front in Figure 1, but here it seems to also refer to the upper depths of permafrost. Line 32: QIT needs to be better defined as the sum of latent + sensible heat flux into the frozen soil below the thaw front. p.3 Line 3: dSTL could be defined more straightforwardly as the heat storage in the active layer Line 7: replace "melting" by "soil thawing" Line 16-18: Please reformulate and clarify p.4 Line 5-6 : Stress that water tracks are linear features Line 15: how was CG determined? Were constant moisture conditions assumed between wet and dry soils? Line 17: Important to state late-summer conditions, as it is the only reason why constant thaw depths can be used. Line 19-20: Please reformulate Line 20 : Physiographic descriptions are lacking for the sites. Slopes and slope aspects are important elements for polar locations, and any difference in aspect and angle can strongly influence timing and magnitude of solar radiation. Looking at the shading and water track orientation in Figure 2, it seems as if slope aspects are not identical between sites. If slopes angles are low, this might not be a big issue, but it requires clarifications. p.5 Figure 2: It could be more useful to have general map of TV, with a single point to designate the study sites, and pictures of the field sites. Figure 3 could also be made smaller and included in it. Line 1: The ice table is a 1.9°C? How is this logical? Shouldn't it be assumed that the ice table is at 0°C as is the case in the water track? Line 3: How was this measured? What are

the values used for the water track and the reference site? This is important for your modelling, and values should be given. p.6 Line 17: Levy et al. 2011 say that surface darkening occurs on 1-3 m, yet a width of 10 m is used. Why? Results: It would have been useful to have access to the meteorological data and ground temperature data, either as appendices or supplemental material, or even to show them as results instead of Figure 3, which could be included in Figure 2. Line 22: How much smaller? Please state with %, maybe mean % and standard deviation. In addition, appendix B shows that Q*s isn't really smaller in one of two instances. Line 28-29: Here QIT is defined as the energy used to melt (sic) permafrost. Clearly 5.2 Mj wasn't used to further lower the thaw front, or it would have moved significantly. In fact, QIT includes both the energy transferred to permafrost as sensible heat and the latent heat used to thaw permafrost (or to melt the ice in permafrost). It could be said this energy is used to warm and thaw permafrost. p.7 Figure 3: The scale is too small for what is actually shown. It could be smaller and included in figure 2. What are the density lines showing? Density of water track contribution? If so, the % seem to be inverted as your smaller area only has 50 % of water track contribution. Line 4-5: Does this timeline correspond to max solar radiation if you correct for slope aspect? Line 4-5: Albedo was stated to be 0.15 in water tracks and 0.22 in non-water tracks soils (Levy et al. 2013). This is the kind of comparison that could be discussed. Line 6: What is the surface temperature? Please provide data. p.8 Figure 4 could benefit from showing totals partitioned between references and periods, as a cumulative histogram. Line 1: Figure 4 shows total, and QH is reduced to 0.8 in water tracks, not 0.7. Line 1-2: Add reference to Figure 5 Line 10: This section could benefit from links to the existing literature, as active layer depths are known for water tracks and non-water tracks in the area. Line 20: How was thermal conductivity measured? It would be interesting to quantify the respective roles of increased energy input and thermal conductivity in the daily energy budget. p. 9 Line 6: It is suggested that energy travels more rapidly toward permafrost in the water track, yet this doesn't appear clearly in Figure 6. As Qs* increases in the reference, so does the active layer temperature (dSTL), with permafrost heat flux (QIT) following closely. In the water track, this latter heat flux seems delayed by about 18 hours. Otherwise, how could heat flux toward permafrost occur before the soil even begins to warm in a downward process? Please clarify. Line 7: Why are there citations at the end of a question? Line 10: These scenarios should include the same parameters (precipitations, insolation, temperature). The first is not a climate scenario, rather an arbitrary 50 % increase in water track abundance. It is not clear how this could occur, as it would require new snowpatches locations. A simpler, more straightforward approach would be to determine if the future would be wetter or drier. This could be done using increases-decreases in area % of water track surfaces, and computing the respectful SEB components for each increment. p.10 Figure 6 caption: Negative energy fluxes. . . These do not appear except for dSTL, so this sentence could be removed. The following sentence could specify how out of the thawed layer (aka the active layer) is both into permafrost (QIT) and toward the atmosphere as QLE or QH. Line 4: Again, how would increase snow melt increase water track abundance? This suggests increased precipitations and new snowpatches. Line 5: a total of 4.4% of what? p.11 Line 1: Increased solar radiation will create a feedback that would decrease solar radiation? Does this mean that no increase in solar radiation is possible in the Dry Valleys? Line 10-11: Please reformulate Line 16: This is the central message of the paper, and should be what is put forward in the abstract and what the introduction leads to. The climate change aspects are secondary to this scientific finding, and are not as sound as this sentence is. Line 17: respond faster. . . Why? It seems as if water tracks as hydrological features are resilient to change, and might even benefit from warmer temperatures. p.14 Table C1: The second row of Water Track is redundant. The first row could simply say 26/12 to 21/01. Otherwise please explain in the caption.

Technical corrections: General comment: Whenever possible, please abstain from using abbreviations, except for long terms which appear often. For example, eddy-covariance could be written in the long form throughout the text. p.1 Line 4: water track instead of water-track. Please correct all other occurrences. Line 6: state-of-the-art

is used a few times in the manuscript. I would suggest removing this, as it tends to age poorly. Please remove all other occurrences. p.2 Line 2: thermokarst p.3 Line 5: Please define CG and z here Line 18: Please define T and q here. This sentence would benefit being re-written and broken down. p.5 Line 10: corrected instead of correction Line 12: was also applied p.6 Line 15: replace wet water-track soils by water track p.7 Figure 3 caption: replace Eddy-Covariance by eddy-covariance Line 5: at the water track was can be explained p.8 Line 14: lower case r in Reference p.11 Line 8: replace an increase in by greater Line 19: by ither

---

## Referee Comment (RC2) · Becky Ball (Referee) · 14 Feb 2019

The authors present a succinct, clearly written analysis of energy fluxes from water tracks in the McMurdo Dry Valleys of Antarctica, in comparison to dry soils that cover most of the landscape, in an effort to foreshadow future changes that might come with increased frequency or extent of water tracks. The methodology used to measure energy balance appears justified and sound. My expertise does not lie in eddy covariance, so I will leave analysis of that aspect to other reviewers. From the perspective of an ecosystem ecologist/biogeochemist who has previously worked on these ecosystems, the manuscript currently reads as one that will be of interest to other scientists

studying either the impacts that water tracks can have on their surrounding ecosystem, or the overall energy balance of this dry valley ecosystem. That makes the publication as-is a very useful publication to a particular audience. The authors might care to think about how they could broaden its impact by either applying their results to an improved understanding of the water track as an ecosystem, or perhaps by proposing some testable hypotheses that could follow from the differences measured in the relative importance of particular energy fluxes. For example, the ground measurements are a snapshot during a field season, so are there hypotheses that could be posed about how this would scale up to more frequent or new or larger water tracks that could then be tested? How might the energy balance difference relate to differences measured in the biologic/geochemistry of water tracks? That might extend beyond the reach of the data presented, but perhaps it could be also be posed in the form of hypotheses. Even my published work with co-author Levy is now old enough to be beyond TC's definition of conflict of interest, so it could be interesting to have continued thought put towards the connections between the physical and biological characteristics of water tracks, which tend to be studied and reported separately to different scientific audiences.

Minor comments: In the last paragraph of section 4, a 10m width for water tracks and 20 m width for streams is specified. Is there by chance published data that could be cited to bolster this? I don't really argue the size classifications, but a justification might be useful. In the last sentence of the conclusion: I assume you mean "either" not "ither".

---

## Referee Comment (RC3) · Anonymous Referee #3 · 27 Feb 2019

This manuscript presents a comparison of surface energy balance for a water track and for two reference locations in the Taylor Valley during 26 days in summer of 2012-2013. The main aim of the study was to evaluate the hypothesis that water tracks alter the surface energy budget, mainly due to the higher moisture content of the soil. The energy balance estimation relied on eddy-covariance (EC) method, net radiation measurements and soil temperature profiles.

I am not an expert on these cold ecosystems, yet I do have some experience on the EC method and will concentrate on EC related issues in my review. It seems that the other two reviewers are experts on the Antarctic ecosystems and hence our fields of

expertise complement each other.

In general the paper is well-written and it has a clear structure which is easy to follow. The available data set is not large, only 26 days, and hence far reaching conclusions cannot be drawn based on it. Nevertheless, the authors do their best and it should be also acknowledged that this is the first EC study in this remote location. However, there is one major flaw in this study which is related to forcing the energy balance to be closed. This should be changed before the manuscript can be accepted for publication. Please see more details below. In my view the manuscript is otherwise good and hence I recommend accepting it for publication after minor revisions suggested below, in addition to the modifications suggested by the other two reviewers.

SPECIFIC COMMENTS

1) The authors estimated QIT (the energy input to the permafrost) as a residual of the surface energy balance, meaning that they force the energy balance to be closed. However, there is a plethora of publications out there that show that at EC sites the energy balance is almost never closed, meaning that incoming energy exceeds the sum of outgoing energy and energy stored in the system (see e.g. Reed et al., 2018; Stoy et al., 2013; Hendriks Franssen et al., 2010; Leuning et al., 2012). Usually 10-30 % of the energy is missing. It is still unclear why the energy balance is not typically closed at EC sites, yet it is often hypothesized to be related to sampling mismatch between instruments, instrumental problems, under sampled large eddies transporting heat, terrain heterogeneity or energy storage in soils, air and biomass below the measurement height (Wilson et al., 2002; Leuning et al., 2012). The reasons might also be different for different sites. Be that as it may, this issue should be acknowledged also in this manuscript in question. Hence, I argue that the energy balance residual cannot be assumed to be equal to QIT and the authors should analyze it only as energy balance residual, not some specific real energy flux. Furthermore, if QIT is related to energy input to the permafrost shouldn't it be directed downwards (i.e. it should be negative) since melting ice requires energy? Now the QIT estimated from the residual is mostly

positive which seems physically implausible. Also Anonymous referee #1 pointed out some unphysical behavior of the estimated QIT when compared with dSTL which might stem from the fact that the energy balance residual is not equal to the real QIT.

2) The measurement campaign lasted only 26 days and hence far-reaching conclusions cannot be based on it. This is something that should be discussed in the text. I mean, there are likely seasonal changes in the energy fluxes in both areas, reference and water tracks, and these seasonal changes are not necessarily the same. The authors should discuss what part of the summer these measurements likely represent well and how the energy fluxes are likely changing during the summer.

MINOR COMMENTS

page 1 line 4 To me "during the Antarctic summer of 2012-2013" sounds like that the measurements lasted for several months. Please explicitly mention the length of the measurement campaign (26 days) here.

p. 4 Section 3 More information about the EC setup is needed. What was the sonic anemometer measurement height? Was the gas analyser next to the sonic or below it? What was the horizontal and vertical separation between the two sensors? Were the EC instruments clearly above the surface roughness elements (e.g. rocks)? How rough was the surface in the three locations? Give estimates e.g. for the roughness lengths

p. 5 Section 4 Did you filter out the low turbulence periods from the EC data? During low turbulence EC is not measuring accurately the surface gas exchange and hence these periods should be filtered out e.g. by removing periods with low friction velocity. This applies also to water vapor fluxes. This is somewhat linked to the non-closure of energy balance at EC sites (Wilson et al., 2002)

p. 5 l. 11 Buoyancy correction by Schotanus et al. (1983) is slightly wrong and one should follow van Dijk et al. (2004) instead. Schotanus et al. (1983) is missing a term

0.51<q><Ts'w'> from the right-hand side of their equation (8). Here brackets denote temporal averaging, q air specific humidity (kg kg-1), Ts temperature (K) measured with the sonic anemometer and w vertical wind speed (m s-1).

p. 5 l. 13 Foken et al. (2004) quality flagging scheme gives quality flags between 1 (best quality) and 9 (worst quality). Here you mention that periods with quality flags equal or smaller than 1 are filtered out. Please clarify

p. 6 l. 5-7 It is a bit unclear what was the overall data coverage after filtering. Please clarify and mention it explicitly for each site.

p. 6 l. 11 This 79 % is quite peculiar value. Why not to use nice round number like e.g. 80 %?

p. 6 l. 8-21 and elsewhere Please use consistent naming for the different surface types. For example, here "stream channel" equals "River" in Fig. 3, right?

p. 7 l.12 – p.8 l. 2 It is unclear what is done here. What is the difference between the ratios reported here e.g. for QH? Please rewrite and clarify.

p. 8. l. 3 How much later QH peaked at the water track? Two hours? Please add this information to make it more concrete

TECHNICAL CORRECTIONS

p.4 l. 10 You are referring to Fig. 3 before Fig. 2. You need to go in order.

p. 4 l. 27 You can remove "and carbon dioxide" from the sentence since those measurements were not used in this study.

p. 5 l. 10 Please replace "spectrally correction" with "corrected for low- and high-pass filtering"

p. 5 l. 12 You can remove "and carbon dioxide fluxes" from the sentence since those measurements were not used in this study.
p. 5 l. 9-10 To be exact the method by Wilczak et al. (2001) does not eliminate the mean vertical wind. It tries to eliminate the mean vertical wind caused by tilted sonic anemometer.

p. 6 l. 21 "Each class was assigned an adequate momentum surface roughness length from literature and used in footprint calculations", right?

p. 7 Fig.3 "Glacial Till" and "Ice" cannot be separated from each other in grayscale figure. Please change the colors.

p. 7 l. 5 remove "was" after water track.

REFERENCES

van Dijk, A., Moene, A. F., & de Bruin, H. (2004). The principles of surface flux physics: Theory, practice and description of the ECPACK library.

Hendricks Franssen, H. J., Stöckli, R., Lehner, I., Rotenberg, E., & Seneviratne, S. I. (2010). Energy balance closure of eddy-covariance data: A multisite analysis for European FLUXNET stations. Agricultural and Forest Meteorology, 150(12), 1553–1567. https://doi.org/https://doi.org/10.1016/j.agrformet.2010.08.005

Leuning, R., van Gorsel, E., Massman, W. J., & Isaac, P. R. (2012). Reflections on the surface energy imbalance problem. Agricultural and Forest Meteorology, 156, 65–74. https://doi.org/https://doi.org/10.1016/j.agrformet.2011.12.002

Reed, D. E., Frank, J. M., Ewers, B. E., & Desai, A. R. (2018). Time dependency of eddy covariance site energy balance. Agricultural and Forest Meteorology, 249, 467–478. https://doi.org/https://doi.org/10.1016/j.agrformet.2017.08.008

Schotanus, P., Nieuwstadt, F. T. M., & Debruin, H. A. R. (1983). Temperature-Measurement with a Sonic Anemometer and its Application to Heat and Moisture Fluxes. Boundary-Layer Meteorology, 26(1), 81–93.

Stoy, P. C., Mauder, M., Foken, T., Marcolla, B., Boegh, E., Ibrom, A., et al. (2013). A

data-driven analysis of energy balance closure across FLUXNET research sites: The role of landscape scale heterogeneity. Agricultural and Forest Meteorology, 171–172, 137–152. https://doi.org/https://doi.org/10.1016/j.agrformet.2012.11.004

Wilczak, J. M., Oncley, S. P., & Stage, S. A. (2001). Sonic Anemometer Tilt Correction Algorithms. Boundary-Layer Meteorology, 99(1), 127–150. https://doi.org/10.1023/A:1018966204465

Wilson, K., Goldstein, A., Falge, E., Aubinet, M., Baldocchi, D., Berbigier, P., et al. (2002). Energy balance closure at FLUXNET sites. Agricultural and Forest Meteorology, 113(1–4), 223–243. https://doi.org/http://dx.doi.org/10.1016/S0168-1923(02)00109-0

---

## Short Comment (SC1) · 4 Mar 2019

Thank you for the thoughtful, response–Becky! We think that the energy balance of the water tracks may play a significant role in any temperature or water-activity-related biogeochemical reactions that happen in the tracks. We'll assemble some thoughts on this and will include them in the revisions. You're absolutely correct that this pilot study was a snap-shot in a pair of nearby locations, which makes it a useful initial datapoint for a much larger ecosystem.

---

## Short Comment (SC2) · 4 Mar 2019

Many thanks to reviewer 1 for their thoughtful and thorough comments! We're working on addressing all of the comments and will be back soon.

Your questions about climate change and landscape response trajectories for the MDV are extremely interesting. For many water tracks, snow abundance, distribution, and duration will be an important factor for predicting changes to wetted area. While for others fed by ground ice, deepening active layer thaw is all but guaranteed to produce additional melt. This, in turn, may drive subsidence, which could trap new snow banks, encouraging further water track activity. We'll explore these ideas in more depth in our

formal response.

---

## Author Comment (AC1) · 18 May 2019

Please find the final responses to all three referee comments, the revised manuscript, the revised manuscript with highlighting of changes and the added supplement to the manuscript in the supplement to this author comment.

Best regards, Tobias Linhardt, Joseph Levy and Christoph Thomas

Please also note the supplement to this comment:
https://www.the-cryosphere-discuss.net/tc-2019-8/tc-2019-8-AC1-supplement.zip

---

## Author Response (AR1)

This author response to comments from editor Philip Marsh is structured as follows:

Editor comments

Author response

Dear Dr. Linhardt. I have read the revised manuscript carefully and feel that you have addressed the reviewers comments and have made the necessary, and substantial, changes to your paper.

Dear Philip Marsh, we thank you for acknowledging our efforts for the manuscript revision. We are glad that we have received lots of constructive criticism and suggestions and are now happy with the result too.

I would request that your improve Figure 1 in the supplemental material. Currently the gray scale of this figure shows that most of the terrain has slopes greater than 5 degrees. With large, solid black regions on the map it is extremely difficult for readers to interpret the data the map is showing. With this minor change, I am recommending your paper for publication in The Cryosphere.

We changed the color scale for slopes in supplementary Fig. 1 to shades of gray and green and hope that different slopes are more easily discernible now than with the previous gray scale.

Thank you for taking the time and effort to make substantial changes to your paper.
Phil

Thank you for your supportive handling of the peer-review process!
Tobias, Joe and Chris

On the following pages you find the manuscript with markup of changes relative to the version from the final response to the referee comments.

[revised manuscript text omitted]